# The Associations of Intimate Partner Violence and Non-Partner Sexual Violence with Hypertension in South African Women

**DOI:** 10.3390/ijerph19074026

**Published:** 2022-03-29

**Authors:** Kim Anh Nguyen, Naeemah Abrahams, Rachel Jewkes, Shibe Mhlongo, Soraya Seedat, Bronwyn Myers, Carl Lombard, Claudia Garcia-Moreno, Esnat Chirwa, Andre Pascal Kengne, Nasheeta Peer

**Affiliations:** 1Non-Communicable Diseases Research Unit, South African Medical Research Council, Tygerberg, Cape Town 7505, South Africa; andre.kengne@mrc.ac.za (A.P.K.); nasheeta.peer@mrc.ac.za (N.P.); 2Gender and Health Research Unit, South African Medical Research Council, Cape Town 7505, South Africa; naeemah.abrahams@mrc.ac.za (N.A.); rachel.jewkes@mrc.ac.za (R.J.); shibe.mhlongo@mrc.ac.za (S.M.); esnat.chirwa@mrc.ac.za (E.C.); 3Office of the Executive Scientist, South African Medical Research Council, Cape Town 7505, South Africa; 4SAMRC Unit on the Genomics of Brain Disorders, Department of Psychiatry, Faculty of Medicine and Health Sciences, Stellenbosch University, Stellenbosch 7600, South Africa; sseedat@sun.ac.za; 5Alcohol, Tobacco and Other Drug Research Unit, South African Medical Research Council, Cape Town 7505, South Africa; bronwyn.myers-franchi@curtin.edu.au; 6Curtin enAble Institute, Faculty of Health Sciences, Curtin University, Perth, WA 6102, Australia; 7Biostatistics Unit, South African Medical Research Council, Cape Town 7505, South Africa; carl.lombard@mrc.ac.za; 8UNDP/UNFPA/UNICEF/WHO/World Bank Special Programme of Research, Development and Research Training (HRP), Department of Sexual and Reproductive Health and Research, World Health Organization, 1211 Geneva, Switzerland; garciamorenoc@who.int; 9School of Public Health, University of Witwatersrand, Johannesburg 2193, South Africa; 10Department of Medicine, University of Cape Town, Cape Town 7925, South Africa

**Keywords:** intimate partner violence, non-partner sexual violence, rape exposure, gender-based violence, hypertension

## Abstract

This study describes associations of intimate partner violence (IPV), non-partner sexual violence (NPSV) and sexual harassment (SH) exposures with hypertension in South African women aged 18–40 years. Baseline data (*n* = 1742) from the Rape Impact Cohort Evaluation study, including a history of sexual, physical, emotional and economic IPV, NPSV and SH were examined. Hypertension was based on blood pressure ≥140/90 mmHg or a previous diagnosis. Logistic regressions were adjusted for traditional hypertension risk factors and previous trauma (e.g., recent rape). Hypertension was more prevalent in women with a history of all forms of IPV, NPSV, and SH, all *p* ≤ 0.001, compared to women without. Frequent NPSV (adjusted odds ratio: 1.63; 95% CI: 1.27–2.67) any SH (2.56; 1.60–4.03), frequent physical (1.44; 1.06–1.95) and emotional IPV (1.45; 1.06–1.98), and greater severity of emotional IPV (1.05; 1.02–1.08) were associated with hypertension. Current depression, post-traumatic stress symptoms and/or alcohol binge-drinking completely or partially mediated these associations. This study shows that exposure to gender-based violence is associated with hypertension in young women. Understanding the role of psychological stress arising from abuse may enable the development of prevention and management strategies for hypertension among women with histories of abuse.

## 1. Introduction

Globally, about one-third of women are affected by intimate partner violence (IPV) or non-partner sexual violence (NPSV) [1,2,3]. These human rights violations are a major public health concern as women with a history of abuse often suffer adverse health consequences, ranging from reproductive health issues to mental health issues, such as depression, suicidality, anxiety, and substance use disorders [3,4]. Some studies suggest that IPV and NPSV increase the risk of subsequent cardiovascular disease (CVD) [5,6], including hypertension, via the contribution of psychological stress [7,8]. It has been postulated that acute and chronic stressors may prolong the activation of the hypothalamic-pituitary-adrenal axis and the autonomic nervous system, and contribute to chronic inflammation and atherosclerotic progression [8,9]. These changes lead to increases in blood pressure (BP), glucose and lipid levels, and endothelial dysfunction [10]. Additionally, smoking and heavy alcohol use, both maladaptive coping strategies in response to stress, are commonly reported among abused women and may contribute to CVDs and hypertension [8]. Depressive symptoms are known predictors of CVD, including hypertension, and can persist long after the abuse-related incident has stopped [11,12]. An understanding of these associations and the factors mediating the relationship may have clinical utility and could inform healthcare policies and service delivery for vulnerable women [13].

IPV is the most common form of gender-based violence (GBV) and comprises sexual, physical, emotional or economic abuse. Other common forms of GBV include NPSV (i.e., perpetrators who are not intimate partners) and sexual harassment (SH). Research has shown that the psychosocial impact of GBV commonly differs by the type(s) experienced, their frequency, and the relationship of the victim to the perpetrator (intimate partner vs. non-partner) [14]. This is particularly pertinent in South Africa, a country experiencing a high burden of gender-based abuse, as well as high rates of hypertension.

The 2016 South African Demographic and Health Survey (SADHS) reported that among adult women (≥18 year old), 26% had been exposed to physical, sexual or emotional IPV [15]. However, much higher levels of IPV were reported in smaller but dedicated non-population-based studies (e.g., 65.2% of women aged 18–30 years residing in informal settings reported physical and/or sexual IPV in the past year) [16]. Furthermore, without accounting for under-reporting, self-reported NPSV prevalence was 12–25% in community-based studies [17,18], considerably higher than the global average of 8% [2].

The 2016 SADHS also showed that nearly half (46%) of the women aged 15 and older who participated in the survey had hypertension [15], which was much higher compared with the 32% estimated in much older populations globally [19]. However, to date, the association between these two epidemics in South Africa has not been rigorously examined [20]. Understanding the relationship of IPV, NPSV, and SH with hypertension is important for the prevention and management of hypertension. An opportunity to explore this relationship was presented by the Rape Impact Cohort Evaluation study (RICE), where both biological and behavioural data were collected from two groups of participants and where it was possible to examine the associations of various types and frequencies of abuse by intimate- and non-partners with hypertension, as well as the mediating effects of potential hypertension risk factors. This study therefore aimed to investigate the associations of IPV, NPSV and SH exposures with hypertension, and the mediating effects of potential hypertension risk factors on these associations among South African women aged 18 to 40 years.

## 2. Material and Methods

### 2.1. Study Designs and Populations

This study used the baseline data of the longitudinal Rape Impact Cohort Evaluation (RICE) study, which was conducted in the Greater Durban Metro area, KwaZulu-Natal Province, South Africa. The study recruited 1799 participants who recently reported a rape (rape exposed participants) (*n* = 852) from five public-health-based post-rape service centres, while control participants (non-rape exposed) (*n* = 947) were recruited from waiting rooms of public primary healthcare clinics in the same locality as the rape service centres. The study protocol, published previously, describes the detailed methods [21]. In brief, women who presented to the sexual assault service centres following a rape incident were invited to participate in the study with the baseline interviews conducted within 20 days of the incident rape event. The South African Medical Research Council Ethics Committee provided ethical approval for this study (SAMRC; EC019-10/2013), and all participants provided informed consent.

### 2.2. Data Collection

Baseline data were collected between August 2014 and March 2019 using an electronic data capturing and management system (Bryant System), where the data were captured on electronic case report forms which were available on personal digital assistants (PDAs), with built-in checks for quality control. Data were encrypted at the point of collection and sent via the internet to a dedicated server, from which it was further checked, downloaded, and stored for future use. The fieldworkers administered validated questionnaires and clinical data were collected by research nurses.

### 2.3. Measurements and Definitions

#### 2.3.1. Clinical and Biochemical Parameters

Standardised techniques were used for anthropometric measurements (height, weight). Body mass index (BMI) was calculated as weight in kilograms (kg) divided by height in metres-squared (m^2^), and BMI ≥ 25 kg/m^2^ defined overweight and obesity.

BP was measured using a digital BP monitor (Omron, M6 Comfort, Hoofddorp, The Netherland) after the participant was seated in a resting position for at least five minutes. Three measurements were taken three minutes apart, and the average of the second and third readings were used in the analysis. Hypertension (binary variable) was defined as systolic BP ≥ 140 mmHg and/or diastolic BP ≥ 90 mmHg or self-reported previously diagnosed hypertension [22]. The latter defined participants who answered in the affirmative to “being told by a doctor or other health worker that you have raised BP or hypertension” regardless of hypertension treatment status.

Biochemical measurements were performed at an accredited pathology laboratory. HIV serostatus was assessed in accordance with WHO guidelines for HIV diagnosis in high-prevalence settings [23]. An initial rapid test was done with Alere Determine Rapid (Abbott laboratories, Matsudo Shi, Chiba-ken, Japan), with a confirmatory test using the FDA-approved Uni-Gold Recombigen HIV1/2 test (Trinity Biotech PLC, Bray, Co.Wicklow, Ireland). A confirmatory test was undertaken on all the positive samples using the combined Abbot IMX ELISA (Abbot laboratories, Johannesburg, South Africa and the Vironosktika HIV1/2 ELISA.

#### 2.3.2. Lifestyle Characteristics

Binge-drinking was defined as consuming ≥ 5 units of alcohol (one unit drink equivalent to one can/bottle of beer, cider, cooler/glass of wine/tot of spirit) on a single occasion within a month. Women who “currently smoke any tobacco products such as cigarettes, cigars or pipes” were defined as current smokers.

#### 2.3.3. Types of Abuse

##### Abuse in Adulthood (≥18 Years)

The analysis in this paper considered that the exclusion criterion for the control group was never having experienced a rape (intimate partner or non-partner rape). The study assessed ever having experienced IPV, including sexual, physical, emotional, and economic IPV, and assessed participants’ perception of partner control. We also assessed NPSV (rape and attempted rape) and SH.

Lifetime intimate partner violence: IPV was assessed using the WHO multi-country study questionnaire [4], which was previously validated in South Africa by Jewkes et al. [24]. These included four items on sexual abuse, five items on physical abuse, seven items on emotional abuse and four items on economic abuse by a previous and/or current husband or partner. Each IPV type was assessed as yes/no and overall IPV was defined as exposure to any of the four types of IPV and categorized as yes/no. The frequency of exposure was dichotomised into never or once (score 0) and more than once (score 1). Exposure to multiple IPV types was classified in three levels: none, 1–2 types, and 3–4 types. The sum of the frequency of exposure for all four IPV types described the cumulative severity (score range from 20 to 80).

Lifetime non-partner sexual violence exposure: Four questions from Jewkes et al. [24] explored experiences of sexual violence by non-partners (three questions about rape experiences and one question on attempted rape), prior to a recent rape incident for those in the rape exposed cohort. An affirmative response to any of the four questions defined sexual abuse by a non-partner. Frequency of exposure was categorised in three levels: never, once, and more than once.

Sexual harassment: A total of nine questions [24] examined having ever experienced unwelcome, inappropriate sexual advances and propositions or threats or coercion to have sex by an employer, a colleague, teacher, principal, lecturer, traditional healer, contractor, taxi driver, and so forth. For example, “*Has any man ever hinted or threatened that you could lose your job, or that your work might be hurt if you did not have sex with him?*” An answer of “yes” to any of the nine questions defined exposure to SH.

##### Abuse in Childhood (<18 Years)

Childhood abuse (CA) was assessed using the Childhood Trauma Questionnaire Short-Form (CTQ-SF) scale [25,26]. Thirteen items were used to measure sexual, physical and emotional abuse and parental neglect before 18 years of age. Frequency of CA was evaluated as never (score 0), some (score 1) and often (score 2). The sum of the exposure frequency for the four types of CA described the cumulative severity. The scores ranged from 0 (no abuse) to 8 (severe or frequent abuse).

##### Exposures to Other Prior Traumatic Events

An adapted Life Event Checklist assessed exposure to 10 traumatic events (e.g., serious injury, being close to death, witnessing the murder of a family or friend, the unnatural death of a family or friend, witnessing the murder of a stranger, torture, being robbed or carjacked at gun- or knife-point, and being kidnapped) [27]. The response for each of the 10 items was scored with a yes (score 1)/no (score 0), and scores for all items were summed up. A higher score was indicative of exposure to more types of traumatic events.

##### Mental Health Issues

Post-traumatic stress symptoms: The validated 34-item Davidson Trauma self-rating scale was used to assess post-traumatic stress symptoms (PTSS) over the previous week [21]. Responses were evaluated on a 5-point Likert scale with scores summed for an overall PTSS score, with higher scores indicating more severe symptoms. Since the definition of posttraumatic stress disorder (PTSD) includes the presence of traumatic stress responses beyond 4 weeks, the stress reactions measured at baseline of the RICE study, that is, in the previous week only, cannot be described as PTSD but as PTSS.

Depressive symptoms: Experiences of depressive symptoms in the previous week were estimated using the 20-item Center for Epidemiologic Studies Depression (CESD) self-report measure [28]. Responses were assessed on a 4-point Likert scale with higher total scores indicating more severe depressive symptoms.

### 2.4. Statistical Analyses

Only baseline data from the RICE study was used for analysis, aided by R statistical software version 3.6.0 (26 April 2019). All participants (i.e., rape-exposed and non-rape exposed groups) were pooled for these analyses. Categorical variables were summarized as count (percentages). The associations of lifetime IPV NPSV and SH with hypertension were examined using multiple logistic regression analyses with adjustments for potential confounders. These included the traditional hypertension risk factors (age, BMI, smoking, binge drinking) and possible risk factors, such as the recent rape exposure, other traumatic exposures (including CA), and HIV infection. We did not find associations between dyslipidaemia and dysglycaemia with hypertension in our prior analysis (data not shown), and therefore did not adjust for these variables in the present regression model. The outcome hypertension was a binary variable; the predictors were various IPV, NPSV and SH experiences and were entered separately in different models, with each model adjusted for the same potential confounders.

The mediation effects of different variables on the association of IPV, NPSV and SH with hypertension were explored using multiple mediation models adjusted for age and recent rape exposure, using the *Laavan* package (http://CRAN.R-project.org/package=lavaan accessed on 17 September 2021) [29]. The potential mediators included BMI, HIV infection, current smoking, binge-drinking, depressive symptoms, and PTSS. Indirect, direct, and total effect estimates (probit regression coefficients) and the proportion of the mediation effect (indirect effect/total effect) were computed, with the significance of the mediation effect tested via bootstrap methods, based on 5000 replications (in simple mediation models only). A *p*-value < 0.05 was considered a statistically significant mediation effect. Full or complete mediation is present when the total and indirect effects are significant, while the direct effect is non-significant. Partial mediation occurs when the total and indirect effects are significant, and the direct effect remains significant.

## 3. Results

### 3.1. Lifetime IPV by Hypertension Status

The RICE study enrolled 1799 participants, but 57 were excluded from this analysis, as they had incomplete data on the variables of interest. Therefore, the final sample consisted of 1742 women with a median age of 24 years (25–75th: 21–29 years). The majority of the women had completed secondary education (8–12 years of education, 89%), were unemployed (80%) and resided in a formal urban area (72%) (Table A1 in Appendix A). Overall, among the women included, 62% (*n* = 1084) reported having experienced any IPV, [sexual IPV (17%), physical IPV (49%), emotional IPV (47%) and economic IPV (19%)] (Table 1 and Table 2). Nineteen percent (*n* = 328) and 7% (*n* = 124) of the women reported lifetime exposure to NPSV and SH, respectively.

Among the women included, 218 (13%) had hypertension (Table 1). Hypertension rates were higher in women with, compared to women without, a history of all forms of IPV (15% vs. 9%), sexual (19% vs. 11%), physical (16% vs. 9%), emotional (17% vs. 9%) and economic IPV (19% vs. 11%). Across the four IPV types, the prevalence of hypertension was always significantly higher in women with exposures to 1–2 types (13% vs. 9%) and 3–4 types (18% vs. 9%) of IPV compared to those without IPV (*p* < 0.001). Similarly, hypertension prevalence was higher in women with, compared to women without, exposure to NPSV (18% vs. 11%) and SH (29% vs. 11%), all *p* ≤ 0.001.

The prevalence of IPV, NPSV and SH by hypertension status are presented in Table 2. The proportion of women who had experienced any IPV was higher in women with, than in those without, hypertension (*p* < 0.001). The prevalence of all IPV types was significantly higher in women with hypertension compared to those without hypertension. Across the four IPV types, the prevalence of exposures to 1–2 types (44% vs. 41%) and 3–4 types (31% vs. 20%) of IPV were significantly higher in women with, compared to those without, hypertension (*p* < 0.001).

### 3.2. Non-Partner Sexual Violence and Sexual Harassment by Hypertension Status

Over a quarter of women (27%) with hypertension reported NPSV compared with 18% of those without hypertension (*p* < 0.001) (Table 2). Furthermore, multiple exposures of NPSV reported at baseline (excluding the incident rape event) was significantly higher in women with hypertension (14%) than in those without (6%). SH was significantly higher in women with, compared to those without, hypertension (17% vs. 6%, *p* < 0.001).

### 3.3. Associations of IPV with Hypertension

Table 3 presents the unadjusted and adjusted associations of lifetime IPV with hypertension. Any exposure to IPV (adjusted odds ratio (aOR): 1.53; 95% CI: 1.10–2.16; *p* = 0.012), physical IPV (OR: 1.44; 95% CI: 1.06–1.96; *p* = 0.021) and emotional IPV (OR: 1.68; 95% CI: 1.23–2.29; *p* = 0.001) were significantly associated with greater odds of having hypertension. Exposure to multiple types of lifetime IPV and a greater frequency of physical and emotional IPV (both *p* = 0.020), but not sexual or economic IPV (*p* ≥ 0.082), were significantly associated with an increased likelihood of hypertension. In addition, the overall severity of IPV and emotional IPV significantly increased the odds of hypertension (*p* ≤ 0.002).

### 3.4. Associations of Non-Partner Sexual Violence and Sexual Harassment with Hypertension

The unadjusted and adjusted associations of NPSV and SH with hypertension are shown in Table 4. Compared with no exposure to NPSV, exposure to more than one episode of NPSV was associated with 63% higher likelihood of hypertension (aOR: 1.63; 95% CI: 1.27–2.67; *p* = 0.020). Exposure to SH (aOR: 2.56; 95% CI: 1.60–4.03; *p* < 0.001) and the cumulative severity of SH were also associated with increased probability of having hypertension (aOR: 1.49; 95% CI: 1.12–1.95; *p* = 0.004).

### 3.5. Mediation Analyses

Results of multiple mediation analysis revealed that the effect of any IPV (*p* = 0.019 for direct effect) with hypertension was partially mediated through depressive symptoms (*p* = 0.032 for mediation effect) and alcohol binge-drinking (*p* = 0.036 for mediation effect) (Table 5). The effect of NPSV with hypertension was fully mediated by depressive symptoms (*p* = 0.119 for direct effect; *p* = 0.020 for mediation effect). The effect of SH with hypertension was partially mediated by depressive symptoms (*p* < 0.001 for direct effect; *p* = 0.041 for mediation effect).

## 4. Discussion

This study found experiences of gender-based violence to be significantly associated with greater odds of hypertension in women. The prevalence of all IPV types was significantly higher in women with hypertension compared to those without hypertension, and demonstrates the potential for serious health consequences in this young sample of abused women. This study also showed how associations with hypertension differed across types, frequency, severity and the relationship with the perpetrator, with a higher frequency of physical, emotional IPV and NPSV significantly associated with hypertension in fully adjusted models. Interestingly, SH was also linked to hypertension, but significant associations were not found for economic and sexual IPV experiences.

Few studies have explored the relationship between GBV and hypertension, and among the few studies, all in developed settings, the variations in abuse measurements makes comparison difficult [30,31]. However, the Nurses’ Health Study II, a longitudinal study, reported an association with emotional IPV but not lifetime physical and sexual IPV after adjusting for many variables, including childhood abuse [30]. Both the USA Behavioral Risk Factor Surveillance Survey (BRFSS) [32] and the large UK retrospective cohort study of 18,547 women reported a non-significant association between physical and/or sexual IPV and hypertension [5]. All of these studies did not assess the frequency or severity of IPV or economic IPV type. Associations between NPSV and hypertension have not been commonly reported either [33]. Research from the US Veteran study of sexual abuse while serving in the military was the only study to report an association between hypertension and NPSV [34].

A key finding in this study was the role of mental health issues such as depressive symptoms, PTSS, and/or alcohol binge-drinking as mediators for the association with hypertension. The mental health impact of rape has been well-described, with risks for PTSS and depression found to be higher in rape survivors compared with other trauma experiences [35,36]. Studies have also differentially reported levels of post-traumatic stress disorder (PTSD) and depression depending on the perpetrator, which implies that different types of sexual violence provoke different reactions in terms of stress [37,38]. This might explain the association of hypertension with NPSV, where stress arises from feelings of shame, guilt, fear of judgement, and being blamed. Rape stigma has been described as having a profound impact on rape survivors’ health [39,40]. Rape stigma strongly intersects with gender norms and poverty and is also strongly linked to previous experiences of trauma, particularly childhood abuse [41,42]. Reducing rape stigma and supporting women is critical to alleviate the negative thoughts and feelings which may prevent help-seeking and healing with the resulting long-term psychological and behavioural sequelae contributing to the development of hypertension [14,43]. Similarly, the significant association of SH with hypertension found in this study, which was mediated by depression symptoms, very likely arises from sexual harassment by perpetrators in powerful positions, such as an employer, senior colleague, respected community member, and so forth, which could create immense stress [41]. Being in a powerless position, together with self-stigmatization (self-blame and shame) may result in survivors feeling trapped, helpless, and reluctant to report the harassment due to fear of being ostracised, victimised, or losing a job. A similar association with sexual harassment was described in a study by Thurston et al. [44] where hypertension was defined as SBP ≥ 130 mmHg or DBP ≥ 80 mmHg in much older women (average age 54 years-old) without a history of hypertension or CVD.

These findings of frequent physical and emotional IPV being associated with hypertension enrich the body of evidence indicating that repeated exposure to violence increases adverse physical and mental health during adulthood [45,46]. It is possible that physical IPV often cannot be hidden from others (i.e., injuries) and this may create more stress in women who do not want others to know about abuse in their relationships, and this may likely lead to psychological distress. The threat of loss of life, exposure to serious injury and not feeling safe in the relationship is likely an ongoing source of stress for women experiencing frequent physical IPV [47]. This severe negative stress response may likely contribute to hypertension in women exposed to violence in intimate relationships [45]. However, the current study did not explore whether seeking assistance and support alleviated some of the stress in women who were exposed to physical IPV. Similarly, frequent emotional IPV was associated with hypertension and is likely related to severe psychological stress imposed by such abuse. Abusive acts like bullying, intimidation, insulting, rejection, isolation, and so forth are approaches often used by abusive partners to dominate or maintain power and control over their female partners [48]. These can induce fear, feelings of helplessness, loss of autonomy, low self-worth and self-esteem and loss of confidence among abused women [14]. However, longitudinal studies in larger and diverse populations are needed to establish causality between abuse against women and the development of hypertension.

Finally, the findings demonstrate that mental health problems, such as depression, PTSS, and binge-drinking, completely or partially mediated the associations of different adult abuse exposures with hypertension. Mental health issues and excessive alcohol and other forms of substance use are common among abused women [49,50,51], and the latter findings underscore the need to address these problems in this vulnerable population. Together with aiding in their healing following abuse exposures, interventions specifically targeting mental health and problematic alcohol use may prevent or delay the development of hypertension among abused women. Psychological support may improve mental and physical health and likely contribute to lowering BP. Evidence-based psychological interventions should be a key component of packages of care for promoting the emotional recovery of abused women. Monitoring changes in the BP of abused women receiving psychological counselling in longitudinal studies will provide insight into whether there are ameliorative effects of these interventions on hypertension. Therefore, it is important that psychological interventions for mental health and harmful alcohol use are among the key interventions offered to women who experience violence and abuse.

## 5. Strengths and Limitations of the Study

The simultaneous exploration of different dimensions of violence exposure (both partner and non-partner exposures) in the same cohort allowed for a better understanding of their comparative impact on hypertension, unlike other studies which often typically assessed a single abuse type or a composite indicator and have focused on IPV [13]. Other strengths of this study include rigorous methodological approaches, such as extensive assessment of abuse exposures using validated questionnaires. The use of standardised objective BP measurements to define hypertension vs. self-reported hypertension allowed for the inclusion of both known and unknown hypertension, and a more accurate assessment of the associations.

Among the weaknesses of the study is the cross-sectional design, which did not allow for inferences on causality. This study included women who reported a recent rape and a control group who reported no previous rape experiences. We controlled for these rape exposures in the analyses. We also controlled for childhood trauma exposures, which have demonstrable links with IPV, NPSV, and adverse cardiovascular, mental health and behavioural outcomes [52]. The study participants were young (median age 24 years, 25–75th: 21–29) and only women, and thus the findings might not be extended to men. Further, these findings are limited to women seeking health care, and may not be generalised to women who are not in care.

## 6. Conclusions

This study highlights the differential associations of the various dimensions (types, frequencies, cumulative severity scores and type of perpetrator) of abuse exposures with hypertension. While primary prevention of violence against women is imperative, there is a need to ensure that women in current abusive relationships are provided with evidence-based interventions for psychological trauma and treated in a nurturing environment without judgement. This will likely allow for quicker healing, minimise the psychological adverse effects of the trauma, and may prevent heavy drinking as a coping strategy. Simultaneously, health professionals need to be aware of both the physical and psychological effects of abuse on women’s health. There is a need to identify, develop and implement effective interventions, including monitoring BP among abused women, not only to overcome the trauma of the abuse, but also to curb the development of hypertension and associated negative mental health sequelae.

## Figures and Tables

**Table 1 ijerph-19-04026-t001:** Prevalence of hypertension by lifetime intimate partner violence, non-partner sexual violence and sexual harassment.

Lifetime Abuse Exposures	Overall (*n* = 1742)	Hypertension (*n* = 218) 12.5%			*p*-Value
	*n*	*n*	%	95% CI	
**Any lifetime IPV**					<0.001
No	658	56	8.5	6.5–11	
Yes	1084	162	14.9	12.9–17.2	
Sexual					<0.001
No	1453	163	11.2	9.7–13.0	
Yes	289	55	19	14.7–24.0	
Physical					<0.001
No	881	83	9.4	7.6–11.6	
Yes	861	135	15.7	13.4–18.3	
Emotional					<0.001
No	920	81	8.8	7.1–10.9	
Yes	822	137	16.7	14.2–19.4	
Economic					<0.001
No	1405	153	10.9	9.3–10.7	
Yes	337	65	19.3	15.3–24.0	
**Lifetime exposure to multiple types of IPV**					<0.001
Nil	658	56	8.5	6.6–11.0	
1–2 type	719	95	13.2	10.9–16.0	
3–4 types	365	67	18.4	14.6–22.8	
**Frequency of lifetime IPV**					
Sexual IPV					<0.001
Never or once	1506	169	11.2	9.7–13.0	
More than once	236	49	20.8	15.9–26.6	
Physical IPV					<0.001
Never or once	1077	104	9.7	8.0–11.6	
More than once	665	114	17.1	14.4–20.3	
Emotional IPV					<0.001
Never or once	1058	100	9.5	7.8–11.4	
More than once	684	118	17.3	14.5–20.3	
Economical IPV					<0.001
Never or once	1486	167	11.3	9.7–13.0	
More than once	256	51	19.9	15.3–25.5	
**Lifetime exposure to NPSV**					0.001
No	1414	159	11.2	9.7–13.0	
Yes	328	59	18	14.1–22.7	
**Frequency of lifetime NPSV**					<0.001
Never	1414	159	11.2	9.7–13.0	
Once	204	29	14.2	9.9–19.9	
More than once	124	30	24.2	17.3–33.1	
**Sexual harassment**					<0.001
No	1618	182	11.3	9.8–12.9	
Yes	124	36	29	21.4–38.0	

IPV, intimate partner violence; NPSV, non-partner sexual violence. *p*-values were from Chi-square tests or Fisher’s exact test, where appropriate.

**Table 2 ijerph-19-04026-t002:** Prevalence of lifetime intimate partner violence, overall and by hypertension status.

Number, (%)	Overall (*n* = 1742)			Hypertension (*n* = 218) 12.5%			No Hypertension (*n* = 1524) 87.5%			*p*-Value
	*n*	%	95% CI	*n*	%	95% CI	*n*	%	95% CI	
**Exposure to IPV: lifetime**										
Sexual	289	16.6	14.8–18.4	55	25.2	19.6–31.5	234	15.4	13.6–17.3	<0.001
Physical	861	49.4	47.1–51.8	135	61.9	55.1–68.3	725	47.6	45.1–50.2	<0.001
Emotional	822	47.2	44.9–49.6	137	62.8	56.1–69.3	685	45.0	42.5–47.5	<0.001
Economical	337	19.4	17.5–21.3	65	29.8	23.8–36.4	272	17.9	16–19.9	<0.001
Any IPV	1084	62.2	59.9–64.5	162	74.3	68–80	922	60.5	58–63	<0.001
**Exposure to multiple types of IPV: lifetime**										<0.001
Nil	658	37.8	35.2–40.4	56	25.7	18.8–33.1	602	39.5	36.8–42.3	
1–2 types	719	41.3	38.7–43.9	95	43.6	36.7–51	624	40.9	38.2–43.7	
3–4 types	365	20.9	18.4–23.6	67	30.7	23.9–38.1	298	19.6	16.9–22.4	
**Frequency of IPV: lifetime**										
Sexual IPV										<0.001
Never or once	1506	86.4	81.7–85.1	169	77.5	71.3–82.8	1337	87.7	85.9–89.3	
More than once	236	13.6	11.9–15.3	49	22.5	17.2–28.7	187	12.3	10.6–14.1	
Physical IPV										<0.001
Never or once	1077	61.8	59.5–64.1	104	47.7	40.9–57.5	972	63.8	61.3–66.2	
More than once	665	38.2	35.7–40.7	114	52.3	45.9–59.6	552	36.2	33.6–38.9	
Emotional IPV										<0.001
Never or once	1058	60.7	58.4–63.0	100	45.9	39.2–52.7	957	62.8	60.1–65.2	
More than once	684	39.3	36.8–41.8	118	54.1	47.7–61.3	567	37.2	34.6–39.9	
Economic IPV										<0.001
Never or once	1486	85.3	83.5–86.9	167	76.6	70.3–81.9	1319	86.5	84.7–88.2	
More than once	256	14.7	12.9–16.6	51	23.4	17.4–29.4	205	13.5	11.6–15.4	
**Lifetime exposure to NPSV**										0.001
No	1414	81.2	79.2–83	159	72.9	66.5–78.7	1255	82.4	80.4–84.3	
Yes	328	18.8	17–20.7	59	27.1	21.3–33.5	269	17.6	15.7–19.6	
**Frequency of lifetime NPSV**										0.001
Never	1414	81.2	79.5–83	159	72.9	67.4–78.8	1255	82.3	80.6–84.2	
Once	205	11.7	10–13.5	29	13.3	7.8–19.2	175	11.5	9.7–13.3	
More than once	123	7.1	5.3–8.9	30	13.8	8.3–19.7	94	6.2	4.3–8	
**Sexual harassment**										<0.001
No	1618	92.8	91.6–94	182	83.5	77.9–88.2	1436	94.2	92.9–95.3	
Yes	124	7.2	6–8.4	36	16.5	11.8–22.1	88	5.8	4.7–7.1	

IPV, intimate partner violence; NPSV, non-partner sexual violence; *p*-values were from Chi-square tests or Fisher’s exact test, where appropriate.

**Table 3 ijerph-19-04026-t003:** Logistic regression analyses (odds ratios, 95% confidence intervals) for the associations of lifetime exposure to intimate partner violence with hypertension.

Exposure Variable	Unadjusted OR (95% CI)	*p*-Value	Adjusted OR (95% CI)	*p*-Value
**Lifetime exposed to IPV**				
Any IPV				
No	1		1	
Yes	1.89 (1.38–2.62)	<0.001	1.53 (1.10–2.16)	0.012
Sexual IPV				
No	1		1	
Yes	1.86 (1.32–2.58)	<0.001	1.40 (0.97–1.99)	0.070
Physical IPV				
No	1		1	
Yes	1.79 (1.34–2.40)	<0.001	1.44 (1.06–1.96)	0.021
Emotional IPV				
No	1		1	
Yes	2.07 (1.55–2.78)	<0.001	1.68 (1.23–2.29)	0.001
Economic IPV				
No	1		1	
Yes	1.95 (1.41–2.67)	<0.001	1.35 (0.94–1.92)	0.093
**Multiple types of lifetime IPV**				
None	1		1	
1–2 types	1.64 (1.16–2.33)	0.005	1.50 (1.05–2.15)	0.026
3–4 types	2.41 (1.65–3.54)	<0.001	1.62 (1.06–2.46)	0.025
**Frequency of lifetime IPV**				
Sexual IPV				
Never or once	1		1	
More than once	2.07 (1.44–2.93)	<0.001	1.40 (0.95–2.05)	0.082
Physical IPV				
Never	1		1	
More than once	1.93 (1.45–2.57)	<0.001	1.44 (1.06–1.95)	0.020
Emotional IPV				
Never or once	1		1	
More than once	1.99 (1.50–2.66)	<0.001	1.45 (1.06–1.98)	0.020
Economical IPV				
Never or once	1			
More than once	1.96 (1.38–2.76)	<0.001	1.23 (0.82–1.80)	0.305
**Cumulative lifetime IPV (score)**				
Overall IPV, range (20–80)	1.04 (1.02–1.05)	<0.001	1.02 (1.01–1.03)	0.002
Sexual IPV, range (4–16)	1.13 (1.06–1.20)	<0.001	1.06 (0.99–1.14)	0.083
Physical IPV, range (5–20)	1.08 (1.04–1.11)	<0.001	1.04 (1.00–1.07)	0.050
Emotional IPV, range (7–28)	1.08 (1.06–1.11)	<0.001	1.05 (1.02–1.08)	<0.001
Economic IPV, range (4–16)	1.17 (1.10–1.24)	<0.001	1.07 (0.99–1.15)	0.062

IPV, intimate partner violence. Logistic regression models adjusted for age, body mass index (BMI), current smoking, binge-alcohol: consuming ≥ 5 units of alcohol on a single occasion within a month, rape-exposed, HIV infection, childhood abuse exposure (scores) and other traumatic exposure (scores).

**Table 4 ijerph-19-04026-t004:** Logistic regression analyses (odds ratios, 95% confidence intervals) for the associations of non-partner sexual violence (NPSV) and sexual harassment with hypertension.

Exposure Variable	Unadjusted OR (95% CI)	*p*-Value	Adjusted OR (95% CI)	*p*-Value
**Lifetime NPSV**				
No	1		1	
Yes	1.74 (1.25–2.39)	0.001	1.26 (0.87–1.83)	0.217
**Frequency of lifetime NPSV**				
Never	1		1	
Once	1.31 (0.84–1.97)	0.218	1.07 (0.67–1.66)	0.777
More than once	2.54 (1.61–3.92)	<0.001	1.63 (1.27–2.67)	0.020
**Lifetime NPSV severity (score), range (4–12)**	1.32 (1.14–1.52)	<0.001	1.11 (0.94–1.31)	0.210
**Ever sexual harassment**				
No	1		1	
Yes	3.22(2.10–4.85)	<0.001	2.56 (1.60–4.03)	<0.001
**Sexual harassment severity (score), range (0–4)**	1.77 (1.38–2.26)	<0.001	1.49 (1.12–1.95)	0.004

IPV, intimate partner violence; NPSV, non-partner sexual violence. Logistic regression models were adjusted for age, body mass index (BMI), current smoking, binge-alcohol: consuming ≥ 5 units of alcohol on a single occasion within a month), rape-exposed, HIV infection, childhood abuse exposure (scores) and other traumatic exposure (scores).

**Table 5 ijerph-19-04026-t005:** Multiple mediation analysis for the effects of lifetime intimate partner violence (IPV), non-partner sexual violence (NPSV) and sexual harassment with hypertension (*n* = 1742).

Confounder/Mediator	Total Effect			Direct Effect			Indirect Effect			Proportion of Mediation Effect
	Co-Efficient	95% CI	*p*-Value	Co-Efficient	95% CI	*p*-Value	Co-Efficient	95% CI	*p*-Value	% (95% CI)
**Any IPV**	0.316	0.145–0.488	<0.001	0.230	0.038–0.421	0.019				
BMI							−0.007	−0.021–0.006	0.282	2.2 (−10–2)
HIV infection							0.010	−0.009–0.030	0.312	3.2 (1–15)
Current smoking							−0.012	−0.072–0.049	0.709	−3.8 (−9–8)
Binge alcohol							0.050	0.003–0.096	0.036	15.8 (1.2–21)
Depressive symptom score							0.028	0.002–0.053	0.032	8.9 (5.2–31.4)
PTSS score							0.017	−0.005–0.039	0.129	5.4 (3–25)
**Any NPSV**	0.267	0.074–0.460	0.007	0.164	−0.042–0.370	0.119				
BMI							−0.005	−0.024–0.014	0.586	−1.9 (−29–1.5)
HIV infection							0.015	−0.013–0.043	0.301	5.6 (−1.5–26)
Current smoking							0.002	−0.019–0.022	0.877	0.8 (−4.3–6.9)
Binge alcohol							0.019	−0.009–0.048	0.180	7.1 (−0.2–17.4)
Depressive symptom score							0.039	0.006–0.072	0.020	14.6 (8–95)
PTSS score							0.034	−0.010–0.077	0.128	12.7 (−2–87)
**Any sexual harassment**	0.656	0.404–0.907	<0.001	0.611	0.348–0.874	<0.001				
BMI							−0.019	−0.051–0.012	0.222	−2.9 (−8.3–0.8)
HIV infection							−0.003	−0.020–0.014	0.729	0.5 (−1.6–4)
Current smoking							−0.003	−0.050–0.044	0.911	0.5 (−8–7.9)
Binge alcohol							0.021	−0.017–0.059	0.282	3.2 (−0.7–5.5)
Depressive symptom score							0.031	0.001–0.060	0.041	4.7 (2–12.4)
PTSS score							0.018	−0.005–0.042	0.129	2.7 (−1–10.7)

Models adjusted for age and recent rape exposure. IPV, intimate partner violence; NPSV, non-partner sexual violence; BMI, body mass index; PTSS, post-traumatic stress symptoms. Binge-alcohol: consuming ≥5 units of alcohol on a single occasion within a month).

## Data Availability

The data used for this analysis are available on reasonable request from the principal investigator.

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
