# Peer review of "The Associations of Intimate Partner Violence and Non-Partner Sexual Violence with Hypertension in South African Women"

_ijerph, 2022, doi:10.3390/ijerph19074026_

Round 1

Reviewer 1 Report

Dear Authors:

 Thank you for the opportunity to review this exciting paper entitled, "The associations of intimate partner violence and non-partner sexual violence with hypertension in South African women. " This manuscript provides new information about the relationship between three types of gender-based violence and hypertension, as well as the mediators between these two variables that can provide health professors some idea to develop and implement effective interventions. Some points need to be revised, and my suggestions are as follows:

Introduction

  1. The authors need to add the aim and hypothesis of this study in the last paragraph.
  2. The title does not match the content of the article. 

Material and methods

  1. Has the IRB reviewed this study?
  2. What is the reliability and validity of all measurements for types of abuse in this study?
  3. Page 4: The authors cannot use the term "mental disease" because PTSS and depressive symptoms are measures of symptoms and cannot be directly said to be a disease.

Results

The form of all tables needs to be modified.

Discussion

Page 11, Line 302: "A key finding in this study was the role of mental disorders such as depressive…" è The higher score of PTSS and depressive symptom cannot prove as a diagnosis of disease. I suggest that the word "mental disorders" needs to be changed.

References

 The format of some references is wrong, such as the name of the Journal needs to be italicized.

Author Response

Dear Authors:

 Thank you for the opportunity to review this exciting paper entitled, "The associations of intimate partner violence and non-partner sexual violence with hypertension in South African women. " This manuscript provides new information about the relationship between three types of gender-based violence and hypertension, as well as the mediators between these two variables that can provide health professors some idea to develop and implement effective interventions. Some points need to be revised, and my suggestions are as follows:

Introduction

  1. The authors need to add the aim and hypothesis of this study in the last paragraph.

    Thank you. We have added to the manuscript this statement ‘This study therefore aimed to investigate the associations of IPV, NPSV and SH exposures with hypertension, and the mediating effects of potential hypertension risk factors on the associations in South African women aged 18 to 40 years.’ Page 2, lines 84-87

  2. The title does not match the content of the article. 

    Thanks for raising this point. We have considered and discussed this, but still of the opinion that the title, ‘The associations of intimate partner violence and non-partner sexual violence with hypertension in South African women’ captures the essence of the manuscript: associations of intimate partner violence and non-partner sexual violence with prevalent hypertension and potential mediators of such associations.

Material and methods

  1. Has the IRB reviewed this study?

    Yes. The RICE study was approved by the South African Medical Research Council Ethics Committee which we stated in the Material and Methods section under sub heading 2.1. The approved protocol had investigation of cardiometabolic risk (the focus of this paper) as a planned objective.

  2. What is the reliability and validity of all measurements for types of abuse in this study?

    In the RICE study, all types of abused were assessed using WHO Multi country study questionnaires which were validated and used previously in multiple South African studies. We have now added a reference in the revised paper. Lines 143, 144, 152,153 and 158.

  3. Page 4: The authors cannot use the term "mental disease" because PTSS and depressive symptoms are measures of symptoms and cannot be directly said to be a disease.Thanks for raising this point. We now refer to ‘mental health issues’ and made the changes throughout the manuscript. 

Results

The form of all tables needs to be modified.

Thanks for raising this point. We have checked and we do not feel that the presentation of our tables violates the guidelines of the journal, and have therefore kept them as they were.

Discussion

Page 11, Line 302: "A key finding in this study was the role of mental disorders such as depressive…" è The higher score of PTSS and depressive symptom cannot prove as a diagnosis of disease. I suggest that the word "mental disorders" needs to be changed.

Thank you. We now refer to mental health issues as stated above

References

 The format of some references is wrong, such as the name of the Journal needs to be italicized.

We have checked the references and it is now done according to ACS citation style as prescribed by the Journal.

Reviewer 2 Report

  1. ABSTRACT

The abstract appears to be well-written and reassumes adequately the major aspects of the present study.

  1. Does the introduction provide sufficient background and include all relevant references?

The introduction seems to present the topic of sexual violence properly, giving a clear overall of the differences between different kind of abuse.  Please consider some minor corrections: Consider introducing the abbreviation NPSV with its full spelling (non-partner sexual violence). Add some literature that shows the particular burden of South Africa in terms of gender-based violence and high rates of hypertension, consider citing some prevalences from other countries to highlight any significant differences. Pg. 2 line 76: include SH in the sentence “Understanding the relationship of IPV and NPSV with hypertension is important for the prevention and management of hypertension”. Consider introducing also the term and prevalence of CA in the introduction, as it is a term used in the methods section.

  1. Is the research design appropriate?

The design of the present study overall seems quite appropriate, considering especially the great number of participants and the various controls in analysis regarding recent rape exposure and mental or physical variables related to BP. In order to enhance clarity, the authors should consider stating more explicitly that only data from baseline of the RICE study was used for analysis.

  1. Methods: Are the methods described in sufficient detail to understand the approach used and are appropriate statistical tests applied?

The authors should consider going providing the total number of participants already in the methods section. Please, explain more in detail how the Bryant System works and how it could have influenced data collection. The authors should define more closely what is meant by “5 units” of alcohol.

Please report which questionnaires drawn from literature were used to assess NPSV and SH, and, if created ad hoc, please provide more examples of items. Please add the number of items of the CTQ-SF, Life Event Checklist and CESD self-report measure.

  1. Results: Are the results clearly presented? Are the results or data that support any conclusions shown directly or otherwise publicly available according to the standards of the field?

Consider reporting descriptive statistics and those regarding the lifetime prevalence of IPV, NPSV and SH as well as hypertension in the total before going into details of Lifetime IPV by hypertension status in section 3.1. Please add the participants’ mean age and consider replacing it with the further details related to the median.

Page 5, line 210: what is meant by “(Data not shown)”, could it be a typo? If not, please make it clearer. There’s another typo on page 5, line 209: please delete “(“.

It could be of interest comparing more closely the groups of those experiencing rape or attempted rape (sample from rape clinic) and those not.

  1. Are the conclusions supported by the results?

The discussion and conclusions appear to be well written, including comparisons with research results of other populations and countries and highlighting the mayor areas of lacking literature. It is to be considered positively that the authors included a section dedicated to strengths and limitations of the study.

  1. Ethics: Does the study’s design, data presentation, and citations comply with standard COPE ethical guidelines?

The study’s design, data presentation and citations comply with the ethical guidelines and the approval from The South African Medical Research Council Research Ethics Committee is present. It is to be considered positively that the Ethic Committee and the number of approval has been reported.

  1. Is the English language and style adequate?

The language is clear overall, but it is advisable to revise punctuation (e.g. Page 3, line 135: “ \ “ should be added between “and” and “or”) and singular and plural forms (e.g. Page 11, line 311 and 312: “intersect” and “experience” must be plural). Moreover, the following changes are suggested:

  • Page 1, line 29: “previous trauma including a recent rape incident” should be substituted by “such as previous trauma (e.g. rape);
  • Page 2, line 47: “health issues to mental ill-health” should be substituted by “to mental health issues”;
  • Page 11, line 307: “some sexual violence induce more or less stress” should be replaced by “that different types of sexual violence provoke different reactions in terms of stress”;

To conclude, it is recommended to find synonyms (e.g. Page 9, line 260: since the word “odds” is frequently repeated throughout the article) and to use impersonal forms instead of the personal ones (e.g. Page 11, line 344: “our” must be replaced by “those”; Page 2, line 79: “we can” should be substituted by “it is possible to”).

  1. Are literature references correct?

The literature references do not conform to the ACS or Chicago styles, both allowed by MDPI journal. For the check I used the “Reference List and Citations Style Guide for MDPI Journals” based on the ACS Style. It is necessary to correct all the list of references and apply the correct style.

  • Check the punctuation in the references list between the author surname and name, just for the name and between the different authors. Until 10 authors you have to report all the names, for more than 10 you can report all the name or just the first 10, then add a semicolon and add “et al.” at the end.
  • Check all the years for the articles, you have to write it in bold.
  • Check all the punctuation between the name of the journal, the year, the volume and page.
  • Check if is it necessary to put [internet] when the journal is online. In the guidelines those are treated like a normal journal.
  • Check if for the articles of online journals, you put the doi and not “available from: http…”.
  • Check if you put all the doi available.
  • Check if is it necessary to put the location of journals’ publication, it doesn’t appear in the references guide lines.
  • Check if you wrote “available online” and not “available from” and then add in brackets the access date.

In particular, check the following citations:

  • World Health Organization (pg. 13, lines 417–423).
  • National Department of Health (NDoH), Statistics South Africa (Stats SA), South African Medical Research Council (SAMRC) and I. (pg. 14, lines 445–448)­ – the last name of the authors isn’t “I” but “ICF”.
  • WHO, LSHTM S. (pg.15, lines 513–514).

Author Response

  1. ABSTRACT

The abstract appears to be well-written and reassumes adequately the major aspects of the present study.

Thank you.

 Does the introduction provide sufficient background and include all relevant references?

The introduction seems to present the topic of sexual violence properly, giving a clear overall of the differences between different kind of abuse.  Please consider some minor corrections:

  • Consider introducing the abbreviation NPSV with its full spelling (non-partner sexual violence).

We have done so on line 45.

  • Add some literature that shows the particular burden of South Africa in terms of gender-based violence and high rates of hypertension, consider citing some prevalences from other countries to highlight any significant differences.

This literature has been added to the revised paper where it reads: ‘Furthermore, without accounting for under-reporting, self-reported NPSV prevalence was 12-25% in community-based studies, and considered higher than the global average of 8%.’ Lines 72-74.

  • Pg. 2 line 76: include SH in the sentence “Understanding the relationship of IPV and NPSV with hypertension is important for the prevention and management of hypertension”.

Thank you. Please see line 78 of the revised paper.

  • Consider introducing also the term and prevalence of CA in the introduction, as it is a term used in the methods section.

As childhood abuse (CA) was not the focus of the paper, we felt that it will probably distract rather if addressed in the introduction. This, however, shouldn’t preclude inclusion of CA as covariates in regression models.

  2. Is the research design appropriate?

The design of the present study overall seems quite appropriate, considering especially the great number of participants and the various controls in analysis regarding recent rape exposure and mental or physical variables related to BP. In order to enhance clarity, the authors should consider stating more explicitly that only data from baseline of the RICE study was used for analysis.

Thank you. We have stated in the first paragraph of method section that ‘This study used the baseline data of the Rape Impact Cohort Evaluation (RICE) study…’. We have further stated in the statistical analysis section that ‘Only data from baseline evaluation of the RICE study was used for analysis using…’. Line 191.

  3. Methods: Are the methods described in sufficient detail to understand the approach used and are appropriate statistical tests applied?

  • The authors should consider going providing the total number of participants already in the methods section.

Suggestion effected. Please see line 92.

  • Please, explain more in detail how the Bryant System works and how it could have influenced data collection.

We have added more details on the Bryant System in the revised paper, lines 104-108.

‘…an electronic data capturing and management system (Bryant System), where the data were captured on electronic case report forms which were available on personal digital assistants (PDAs), with built-in checks for quality control. Data were encrypted at the point of collection and sent via internet network to a dedicated server, from which it was further checked, downloaded, and stored for future use.’

  • The authors should define more closely what is meant by “5 units” of alcohol.

We have added the definition of one unit drink into the revised paper, ‘one unit drink equivalent to one can/bottle of beer, cider, cooler/glass of wine/tot of spirit’. Lines 131-132.

  • Please report which questionnaires drawn from literature were used to assess NPSV and SH, and, if created ad hoc, please provide more examples of items. 

Measures of NPSV and SH used in the RICE study are from Jewkes et al that are cited on page 4, lines 152, 158.

‘Jewkes, R.; Nduna, M.; Levin, J.; Jama, N.; Dunkle, K.; Khuzwayo, N.; Koss, M.; Puren, A.; Wood, K.; Duvvury, N. A Cluster Randomized-Controlled Trial to Determine the Effectiveness of Stepping Stones in  Preventing HIV Infections and Promoting Safer Sexual Behaviour amongst Youth in the Rural Eastern Cape, South Africa: Trial Design, Methods and Baseline Findings. Trop. Med. & Int. Heal.  TM & IH 2006, 11 (1), 3—16. https://doi.org/10.1111/j.1365-3156.2005.01530.x.’

  • Please add the number of items of the CTQ-SF, Life Event Checklist and CESD self-report measure.

We have added to the revised paper, the number of items of the CTQ-SF, Life Event Checklist and CESD self-report measure. Lines 166, 175, and 187.

  4. Results: Are the results clearly presented? Are the results or data that support any conclusions shown directly or otherwise publicly available according to the standards of the field?

  • Consider reporting descriptive statistics and those regarding the lifetime prevalence of IPV, NPSV and SH as well as hypertension in the total before going into details of Lifetime IPV by hypertension status in section 3.1.

We have added these data to the Results section (line 222-226), and in Table 1 of the revised manuscript. We have reported prevalence of hypertension in line 227.

  • Please add the participants’ mean age and consider replacing it with the further details related to the median.

Data for age were skew, hence our choice of median as measure of central tendency.

  • Page 5, line 210: what is meant by “(Data not shown)”, could it be a typo? If not, please make it clearer. There’s another typo on page 5, line 209: please delete “(“.

Thank you. These are typo errors indeed. The corrections are done accordingly. Lines 222, 221.

  • It could be of interest comparing more closely the groups of those experiencing rape or attempted rape (sample from rape clinic) and those not.

Thanks for this important suggestion. However, with the modest number of hypertensive cases among women who reported NPSV (59/328); the study will be very underpowered to reliably perform the comparisons suggested by the reviewer.

  5. Are the conclusions supported by the results?

The discussion and conclusions appear to be well written, including comparisons with research results of other populations and countries and highlighting the mayor areas of lacking literature. It is to be considered positively that the authors included a section dedicated to strengths and limitations of the study.

 Thank you!

  6. Ethics: Does the study’s design, data presentation, and citations comply with standard COPE ethical guidelines?

The study’s design, data presentation and citations comply with the ethical guidelines and the approval from The South African Medical Research Council Research Ethics Committee is present. It is to be considered positively that the Ethic Committee and the number of approval has been reported.

Thank you!

  7. Is the English language and style adequate?

The language is clear overall, but it is advisable to revise punctuation (e.g. Page 3, line 135: “ \ “ should be added between “and” and “or”) and singular and plural forms (e.g. Page 11, line 311 and 312: “intersect” and “experience” must be plural).

Typo errors were corrected throughout. Thank you.

Moreover, the following changes are suggested:

  • Page 1, line 29: “previous trauma including a recent rape incident” should be substituted by “such as previous trauma (e.g. rape);

Done. Line 29 of the revised paper.

  • Page 2, line 47: “health issues to mental ill-health” should be substituted by “to mental health issues”;

Done. Line 47 of the revised paper.

  • Page 11, line 307: “some sexual violence induce more or less stress” should be replaced by “that different types of sexual violence provoke different reactions in terms of stress”;

Done. Line 324-325 of the revised paper

  • To conclude, it is recommended to find synonyms (e.g. Page 9, line 260: since the word “odds” is frequently repeated throughout the article)

Done as suggested. Lines 274, 276 of the revised paper.

  • and to use impersonal forms instead of the personal ones (e.g. Page 11, line 344: “our” must be replaced by “those”;

Done. Lines 343, 362 and 388 and 393 of the revised paper

  • Page 2, line 79: “we can” should be substituted by “it is possible to”).

 Done. Line 81 of the revised paper.

  8. Are literature references correct?

The literature references do not conform to the ACS or Chicago styles, both allowed by MDPI journal. For the check I used the “Reference List and Citations Style Guide for MDPI Journals” based on the ACS Style. It is necessary to correct all the list of references and apply the correct style.

  • Check the punctuation in the references list between the author surname and name, just for the name and between the different authors. Until 10 authors you have to report all the names, for more than 10 you can report all the name or just the first 10, then add a semicolon and add “et al.” at the end.
  • Check all the years for the articles, you have to write it in bold.
  • Check all the punctuation between the name of the journal, the year, the volume and page.
  • Check if is it necessary to put [internet] when the journal is online. In the guidelines those are treated like a normal journal.
  • Check if for the articles of online journals, you put the doi and not “available from: http…”.
  • Check if you put all the doi available.
  • Check if is it necessary to put the location of journals’ publication, it doesn’t appear in the references guide lines.
  • Check if you wrote “available online” and not “available from” and then add in brackets the access date.

We thank the reviewer for the detailed instruction. We have checked and formatted all the references accordingly. The reference list is now in accordance with ACS citation style.

We have corrected three citations pointed out by the reviewer:

In particular, check the following citations:

  • World Health Organization (pg. 13, lines 417–423).

‘Garcia-Moreno, C.; Jansen, HAFM.; Ellsberg, M.; Heise, L.; Watts, C. WHO multi-country study on women’s health and domestic violence against women: Initial results on prevalence, health outcomes and women’s responses; World Health Organization: Geneva, Switzerland, 2005.’ Reference 4, the revised paper

  • National Department of Health (NDoH), Statistics South Africa (Stats SA), South African Medical Research Council (SAMRC) and I. (pg. 14, lines 445–448)­ – the last name of the authors isn’t “I” but “ICF”.

World Health Organization. Global and Regional Estimates of Violence against Women: Prevalence and Health Effects of Intimate Partner Violence and Non-Partner Sexual Violence; World Health Organization: Geneva, Switzerland, 2013.’ Reference 46, the revised paper.

  •  WHO, LSHTM S. (pg.15, lines 513–514).

National Department of Health (NDoH), Statistics South Africa (Stats SA), South African Medical Research Council (SAMRC), and ICF. South Africa Demographic and Health Survey 2016. Pretoria, South Africa, Rockville, Maryland, USA NDoH, Stats SA, SAMRC, ICF. 2019.’. Reference 15, the revised paper
